

# Estimating the Thermodynamic Contribution to Recent Greenland Ice Sheet Surface Mass Loss

Jonathon R. Preece[1], Patrick Alexander[2,3], Thomas L. Mote [1], Gabriel J. Kooperman[1],
Xavier Fettweis [4], and Marco Tedesco [2,3,]

5   [1]Department of Geography, University of Georgia, Athens, 30602, USA.
[2] Lamont-Doherty Earth Observatory, Columbia University, Palisades, 10964, USA.
[3] NASA Goddard Institute for Space Studies, New York, 10025, USA.
[4]Laboratory of Climatology, Department of Geography, SPHERES research unit, University of Liège, Liège, Belgium

10   *Correspondence to*: Jonathon R. Preece (jonathon.preece@uga.edu)

**Abstract.** The Greenland Ice Sheet has become the largest single frozen source of global sea level rise following a pronounced increase in meltwater runoff in recent decades. The pivotal role of anomalous anticyclonic circulation patterns in facilitating this increase has been widely documented; however, this change in atmospheric circulation has coincided with a rapidly warming Arctic. While amplified warming at high latitudes has undoubtedly contributed to trends in Greenland's mass loss, 15   the contribution of this shift in background conditions relative to changes in regional circulation patterns has yet to be quantified. Here, we apply the pseudo-global warming method of dynamical downscaling to estimate the contribution of the change in the thermodynamic background state under global warming to observed Greenland Ice Sheet surface mass loss since the turn of the century. Our analysis demonstrates that, had the recent atmospheric dynamical forcing of the Greenland Ice Sheet occurred under a preindustrial setting, anomalous surface mass loss would have been reduced by over 62% relative to 20   observations. We show that the change in the thermodynamic environment under amplified Arctic warming has augmented melt of the ice sheet via longwave radiative effects accompanying an increase in atmospheric water vapor content. Furthermore, the thermodynamic contribution to surface mass loss over the exceptional melt years of 2012 and 2019 was less than half that of the long-term average, demonstrating a reduced influence during periods of strong synoptic-scale atmospheric forcing.

## 1 Introduction

Greenland Ice Sheet (GrIS) mass loss has rapidly accelerated since the turn of the century (Khan et al., 2015; Kjeldsen et al., 2015; Mouginot et al., 2019; The IMBIE Team et al., 2020; Velicogna et al., 2020), becoming the largest single frozen source of global sea level rise and second largest among all sources after thermal expansion of the warming oceans (Cazenave et al., 2018; Horwath et al., 2022). While highly variable, GrIS mass loss has consistently raised global mean sea level by over 0.5 mm yr$^{-1}$ in recent decades—a rate that outpaces that of the Antarctic Ice Sheet (Smith et al., 2020; The IMBIE team et al., 30   2018; The IMBIE Team et al., 2020) and is approximately equal to that of all other glaciers combined (Cazenave et al., 2018; Zemp et al., 2019). Estimates place the total contribution of Greenland at over 10 mm of sea level rise since the 1990s



(Mouginot et al., 2019; The IMBIE Team et al., 2020). Moreover, most of the impact of recent climate change on GrIS mass balance has yet to be realized as the timescale at which icesheet dynamics adjust to a climate perturbation is an order of magnitude or greater than that of the surface mass balance (SMB) response (Box et al., 2022). Recent work conservatively estimates another ~274 mm of committed sea level rise before the GrIS achieves balance with the current climate state—i.e., even without considering the additional impact of any future warming scenario (Box et al., 2022).

Total mass balance is determined as the SMB—principally, the budget of snow accumulation minus meltwater runoff—less any dynamic loss via solid ice discharge from marine terminating outlet glaciers. Over Greenland, there has been both an acceleration of solid-ice discharge and a decline in SMB over the past few decades (van den Broeke et al., 2009a; Mankoff et al., 2019; The IMBIE Team et al., 2020); however, increased runoff from the GrIS has caused SMB to decline at a rate twice that of the observed increase in dynamic ice loss (Box et al., 2022; Fettweis et al., 2017, 2020; Mote, 2007; Noël et al., 2017). Consequently, SMB reductions have surpassed discharge as the largest source of GrIS mass loss (Mouginot et al., 2019) and, according to global climate models (GCMs) from CMIP5 and CMIP6, SMB losses are expected to exceed mass accumulation on their own by the year 2100 unless the most ambitious mitigation efforts are implemented (Noël et al., 2021).

This stark change in GrIS surface conditions has been associated with a recent shift in summer atmospheric circulation over the North Atlantic. The negative trend in GrIS SMB has coincided with a more persistently negative North Atlantic Oscillation (NAO) and an increase in atmospheric blocking episodes over Greenland (Bevis et al., 2019; Fettweis et al., 2013; Hanna et al., 2015, 2016, 2018b, 2022; Hofer et al., 2017). Indeed, previous studies have shown a significant increase in summer blocking over Greenland since the turn of the century using a variety of blocking detection methods (Davini and D'Andrea, 2020; Hanna et al., 2022; Woollings et al., 2018). Referred to as Greenland blocks, these persistent, anomalous anticyclones have played a key role in encouraging melt of the GrIS via multiple contrasting mechanisms. For example, the positive trend in GrIS surface melt has been linked to Greenland blocking through the suppression of cloud cover by large-scale subsidence within the blocking ridge (Hofer et al., 2017). This reduction in cloud cover has allowed for anomalously high incoming shortwave radiation over the southern ice sheet which, owing to its lower surface albedo, is more sensitive to changes in shortwave radiation than other regions of the GrIS (Hofer et al., 2017; Wang et al., 2019). Other studies, however, have demonstrated the importance of cloud longwave radiative effects, particularly in regions where albedo is high, such as the northern ice sheet and over the high-elevation accumulation zone (Gallagher et al., 2018; Lenaerts et al., 2019; Noël et al., 2019; Wang et al., 2019). Strong southerly moisture transport upstream of a blocking anticyclone in July 2012 supported the formation of low-level cloud cover that produced melt over the highest elevations of the ice sheet for the first time in over a century (Bennartz et al., 2013; Clausen et al., 1988; Mattingly et al., 2018; Meese et al., 1994; Neff et al., 2014; Nghiem et al., 2012). Additionally, the high-amplitude Omega blocking patterns that have undergone the greatest increase in recent summers deliver moisture farther poleward, generating above-normal downward longwave radiation over the most northern portions of



Greenland (Preece et al., 2022)—conditions which have caused pronounced growth of the ablation zone in the region and spurred a disproportionate increase in runoff from the northern drainages of the ice sheet (Noël et al., 2019).

A natural question is whether this shift in summer circulation may be a symptom of climate change. At the hemispheric scale, there are several theoretical frameworks that postulate a link between changes in the meridional temperature gradient under

Arctic Amplification and more frequent persistent weather extremes (Cohen et al., 2014; Coumou et al., 2018; Francis and Vavrus, 2012), and there is mounting evidence of such a link during summer (Cattiaux et al., 2016; Coumou et al., 2015; Di Capua and Coumou, 2016; Kornhuber and Tamarin-Brodsky, 2021; Vavrus et al., 2017). Focusing on summer Greenland blocking more specifically, Liu et al. (2016) demonstrated a relationship between reduced Arctic sea ice and anticyclonic conditions over Greenland using both observations and modeling. However, not only have GCMs failed to capture the positive

trend in Greenland blocking, they consistently predict a decline in blocking frequency in the region (Delhasse et al., 2021; Hanna et al., 2018a)—a discrepancy that constitutes a critical source of outstanding uncertainty regarding a causal link between anthropogenic climate change and the observed shift in summer circulation over Greenland.

Conversely, the change in the background thermodynamic environment, and its resulting impact on GrIS SMB, represents a

more robust signal of climate change than the potential dynamical response outlined above. Multiple well-documented radiative feedbacks have helped warm the Arctic at four times the global average rate (Pithan and Mauritsen, 2014; Rantanen et al., 2022; Serreze and Barry, 2011). This constitutes a likely contributor to the nonlinear decline in GrIS SMB as surface melt would be expected to increase in frequency and magnitude in a warmer, more humid atmosphere.

While the thermodynamic environment over Greenland is surely influenced by changes occurring more broadly throughout the Arctic, local sea-surface conditions may also play an important role. Specifically, sea ice reductions over adjacent waters could further contribute to elevated temperatures over Greenland through the water vapor feedback, wherein a warmer atmosphere together with an ice-free ocean increases atmospheric water vapor, which then enhances longwave radiative forcing at the surface (Pithan and Mauritsen, 2014). Thus, one of the more intuitive ways that sea ice loss could impact the

GrIS is through advection of warm, moisture-enriched air from the neighboring seas. In investigating such a link, studies have revealed a relationship between changes in sea-ice concentration near Greenland and GrIS SMB (Pedersen and Christensen, 2019; Rennermalm et al., 2009); however, these studies fail to separate direct marine influence from any indirect affects that might occur via alteration of the large-scale circulation by oceanic thermal forcing (Ballinger et al., 2019; Liu et al., 2016) or relationships that might arise as a byproduct of mutual forcing of local sea ice concentration (SIC) and GrIS melt by the large-

scale synoptic setting (Ballinger et al., 2018; Stroeve et al., 2017).

Several efforts have demonstrated a minimal contribution from local SIC and SST anomalies during summers of pronounced GrIS melt (Hanna et al., 2009, 2014; Noël et al., 2014). Modeling work has provided one explanation for the lack of marine



influence by demonstrating that persistent katabatic outflow over the ice sheet acts as a barrier to onshore advection (Hanna et
al., 2014; Noël et al., 2014); however, observational evidence suggests that local sea-surface conditions may play an important
role earlier in the Spring preceding GrIS melt events . While melt events during summer and fall are primarily a product of
large-scale atmospheric conditions (Ballinger et al., 2019; Hermann et al., 2020; Noël et al., 2014), recent work has
demonstrated elevated atmospheric moisture and enhanced downwelling longwave radiation over the ice sheet approximately
one week following sea ice retreat in the Baffin Bay and Davis Strait in years of early GrIS melt, suggesting that local sea ice
anomalies precondition the ice sheet for early melt onset (Stroeve et al., 2017).

Disentangling the relative contributions of atmospheric dynamics versus thermodynamics is an intractable problem using
observations alone. Previous studies have utilized regional climate models (RCMs) to examine the sensitivity of GrIS SMB to
perturbations in sea-surface conditions (Hanna et al., 2009, 2014; Noël et al., 2014) or atmospheric thermodynamic fields
(Delhasse et al., 2018); however, none of these efforts examined the combined influence of both atmospheric and sea-surface
conditions. Furthermore, these studies either applied arbitrary perturbations to the targeted boundary fields or only examined
a single melt season and, therefore, did not aim to measure the existent contribution of observed changes in these fields to GrIS
mass loss since its acceleration around the turn of the century.

Here, we provide a more systematic estimate of the relative contributions of dynamical versus thermodynamic change to recent
GrIS surface mass loss using the pseudo-global warming (PGW) method of dynamical downscaling. The PGW method uses
adjusted reanalysis data for the initial and lateral boundary conditions of an RCM (Kawase et al., 2008; Kimura and Kitoh,
2007; Schär et al., 1996). To obtain the adjusted boundary conditions, this method applies a climate change perturbation signal
that is estimated from GCM output by assuming a linear change in the boundary fields between the control period (i.e., the
period of observed reanalysis data) and some alternative period of interest with a contrasting thermodynamic background state
(typically some future period) (Kawase et al., 2009; Rasmussen et al., 2011; Schär et al., 1996). Thus, the PGW technique
effectively isolates the impact of the long-term thermodynamic component of climate change by assuming that the timing and
structure of synoptic disturbances along the RCM's boundaries will be the same in the alternative period as during the control
(Lackmann, 2015).

While the PGW method is typically utilized to simulate future conditions, it can also be used to investigate how recent periods
of climate or individual weather events would have behaved under past conditions. For example, Lackmann (2015) estimated
the thermodynamic contribution of recent climate change to the evolution of Hurricane Sandy by comparing a control run to a
PGW simulation using boundary conditions that were adjusted to reflect the climate of the late 19th Century. Likewise, Kawase
(2008) used a similar approach in a climate change attribution study of the Mei-yu rain band in southern China. Here we use
the PGW method to quantify what the magnitude of GrIS surface melt would have been if the recent dynamical forcing of the
ice sheet had occurred in a preindustrial thermodynamic setting. Specifically, this analysis aims to answer the following





questions: (1) How much of the recent SMB decline can be attributed to local thermodynamic change (i.e., the combined influence of increasing background temperatures and contributions from local sea-surface conditions)? (2) What portion of the thermodynamic influence is due to adjacent SIC / SST change alone? (3) Do sea-surface conditions have a discernible impact on the timing or duration of the GrIS melt season?

## 2. Experimental Design

A model schematic outlining our approach is presented as Figure 1. Atmospheric conditions and GrIS SMB and surface energy balance (SEB) response were modeled using the regional climate model, Modèle Atmosphérique Régional (MAR) (Fettweis et al., 2005, 2020; Lefebre et al., 2005). MAR includes a surface-atmosphere energy and mass transfer scheme with a one-dimensional snowpack model that represents snow grain metamorphism and its impact on albedo, and accounts for the percolation and refreeze of meltwater within the snowpack (Amory et al., 2021; Brun et al., 1989, 1992). For a more detailed description of MAR, we refer the reader to Amory *et al.* (2021). Here, we employed MAR version 3.12 initialized and forced at its lateral boundaries with 6-hourly ERA5 reanalysis data and integrated over a 120x180, 20-km grid with 24 vertical atmospheric levels.

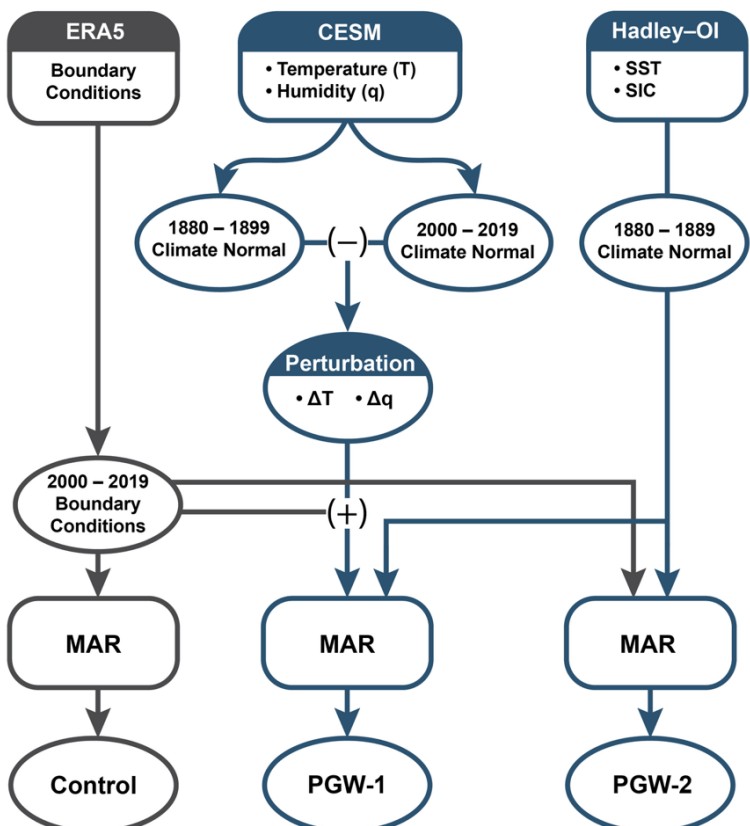

**Figure 1. Model experiment overview.** Model schematic illustrating the design of the control run (gray outlines) and pseudo-global warming experiments (blue outlines). For the control run, all boundary fields including sea surface conditions were sourced from ERA5.



As a control, we forced MAR with ERA5 data spanning the 2000 to 2019 period to provide a representation of historical
conditions during the recent period of anomalous Greenland blocking and the attendant acceleration of GrIS surface melt
(Figure 1, gray components). For the control run, all boundary fields including sea surface conditions were sourced from
ERA5. To simulate the preindustrial thermodynamic state (Figure 1, blue components), we adjusted the boundary conditions
of air temperature and specific humidity using perturbations obtained from the NCAR Community Earth System Model-Large
Ensemble (CESM-LE) project (Kay et al., 2015), while zonal and meridional winds at the model boundaries were left unaltered
to minimize differences in the large-scale atmospheric circulation between the experiment and the control. For the pre-
industrial simulations, we adjusted ERA5 air temperature and specific humidity using a climate change perturbation derived
from the 40 ensemble members of the NCAR CESM-LE project as follows:

$$\Delta x = \overline{x}_p - \overline{x}_c$$


Where $\Delta x$ is the climate change perturbation for variable $x$, $\overline{x}_p$ is the ensemble-averaged, long-term monthly mean of variable
$x$ for a preindustrial reference period of 1880–1899, and $\overline{x}_c$ is the ensemble-averaged, long-term monthly mean of variable x
for a control period of 2000–2019. We then linearly interpolated the monthly climate change perturbations derived from
CESM-LE temporally to a 6-hourly timestep, vertically to ECMWF's L137 hybrid sigma-pressure levels, and horizontally to
the ERA5 grid before adding them to the corresponding ERA5 boundary fields that were used to force MAR.

While GCMs aim to capture the periodicity of internal climate variability, the precise timing of a particular mode of variability
differs between individual ensemble members. Considering a single GCM simulation alone would risk enhancing or
suppressing the magnitude of the climate perturbation signal if the control and alternative periods are characterized by opposing
phases of a relevant mode of internal variability. The same is true for observations. For example, summer atmospheric
circulation has been characterized by an anomalously negative North Atlantic Oscillation (NAO) since the turn of the century—
an atmospheric setting that favors the advection of warm, moist air over Greenland (Fettweis et al., 2013; Hanna et al., 2013;
Henderson et al., 2021; Mattingly et al., 2018; Mote, 1998; Tedesco et al., 2016). Since our intent is to isolate the contribution
of background thermodynamic change to GrIS mass loss, deriving a perturbation signal from observations would overestimate
the baseline change in temperature and humidity around Greenland because it would also include the dynamical contribution
of an anomalously negative NAO during our control period. Taking an ensemble mean of the CESM-LE effectively removes
the noise of internal climate variability by averaging across the differing phases resolved by each ensemble member for a given
date, thereby providing a more appropriate estimate of the change in the mean climate state under global warming.

Figure 2 shows a subset of the monthly surface air temperature and specific humidity perturbation fields that were applied at
the lateral boundaries of MAR. Seasonally, CESM-LE simulates the greatest temperature difference in fall and winter, where



conditions over the surrounding seas were more than 3 °C cooler during the preindustrial period than in the current climate (Fig. 2, top row). The spatial distribution and seasonality of this temperature perturbation signal is consistent with what should be expected under Arctic amplification, which is, in large part, driven by sea ice loss (Screen and Simmonds, 2010).

Differences in surface atmospheric moisture are largely reflective of the Clausius-Clapeyron relation, with drier conditions mirroring locations of cooler temperatures in the preindustrial climate (Fig. 2, bottom row).

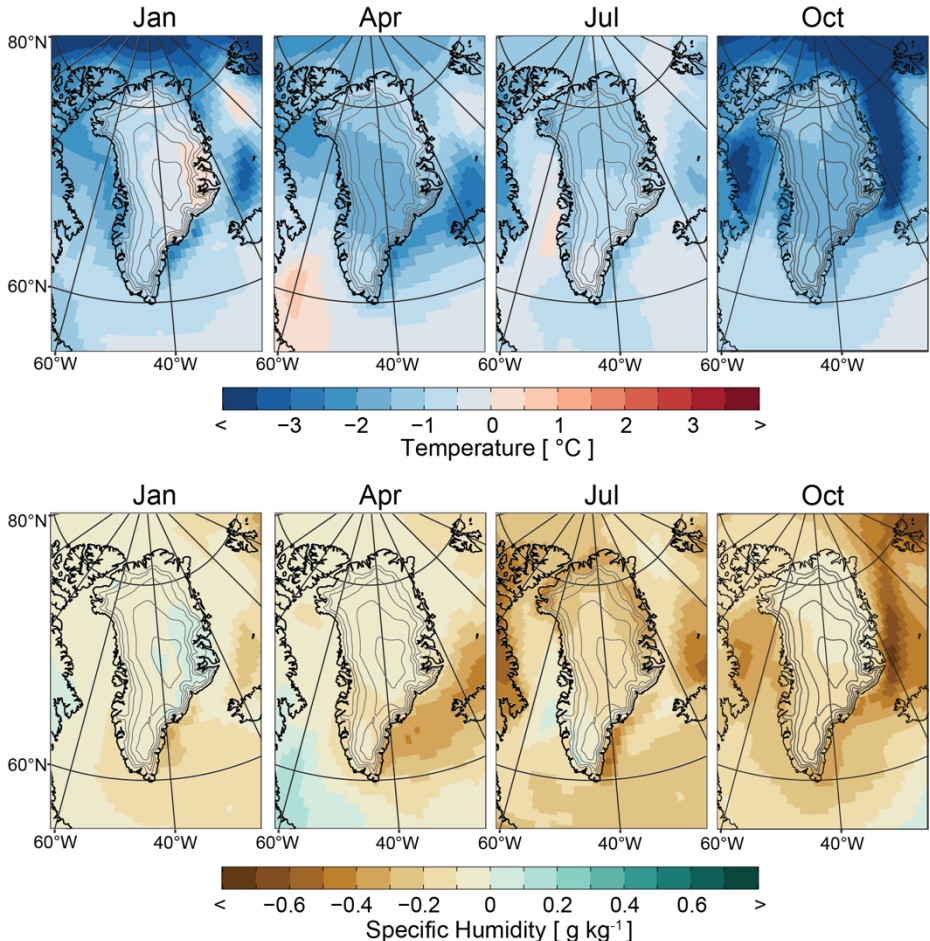

**Figure 2. Climate change perturbation fields.** Perturbation fields derived from CESM-LE for surface air temperature (top row) and specific humidity (bottom row) shown for a selection of months equally spaced throughout the year as labeled at the top of each panel. Perturbation

fields shown for the lowermost model level after vertically interpolating to the same ECMWF L137 hybrid sigma-pressure levels as the ERA5 boundary conditions. Contour interval: 500 m. Range: 1000–3000 m.

The use of a GCM-derived perturbation signal presents issues, however, when dealing with sea ice. The change in sea ice concentration (SIC) is greatest along the sharply defined sea ice front and the GCM's representation may not geographically

align with observations. This misalignment is quite apparent when comparing the perturbation signal to the sub-daily



observations used to force the RCM. Figure 3 presents one such comparison for June 15, 2018, which occurs during a month of exceptionally low SIC in the Greenland Sea. There is a considerable gap between the observed sea ice front and area of greatest SIC change according to CESM-LE, such that the application of this perturbation signal would result in a local minimum in SIC stretching along the original sea ice front, followed by a band of higher SIC stretching from Iceland to
Svalbard that is separated from the main body of sea ice. To avoid this unrealistic circumstance, and to ensure consistency between SIC and sea-surface temperature (SST), we prescribed both SIC and SST in our experimental simulations using 1880–1899 long-term monthly means calculated from the merged Hadley-OI observational dataset (Shea et al., 2020) and interpolated to a 6-hourly timestep.

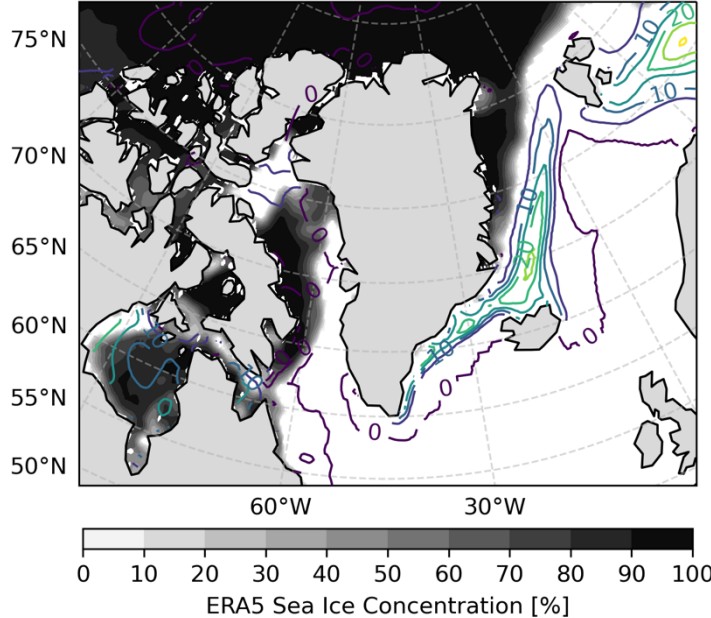

**Figure 3. Sea ice representation.** Comparison of observed sea ice concentration on June 15, 2018 (shading) and the corresponding CESM-
LE climate perturbation signal (contours, 5% interval).

Contrary to global SST trends, there are extensive areas around Greenland where SST during preindustrial period was higher than during the current period—i.e., SST has decreased throughout much of the region since the preindustrial period (Figure S1). This is most apparent during winter and spring when higher preindustrial SST is observed throughout the northern subpolar
gyre to the southeast of Greenland and extending from the southern Greenland coast along the sea ice edge to Svalbard. In summer, SST throughout much of the region was lower during preindustrial period (Figure S1). The spatial and seasonal pattern of lower SST since the preindustrial period matches the fingerprint of the so-called North Atlantic warming hole—an observed decrease in subpolar North Atlantic SST that has been attributed to a weakening of the Atlantic meridional overturning circulation and associated poleward oceanic heat transport as a consequence of global climate change (Caesar et
al., 2018).





After interpolating the Hadley-OI fields to a 6-hourly timestep, we applied the following adjustments based on the work of Hurrell *et al*. (2008) to further ensure consistency between SST and SIC:

- If an interpolated grid cell had a SIC > 90%, we set the SST of that cell to the sea ice freezing point of -1.8 °C.
- Where 15% < SIC < 90% we adjusted SST as follows:

$$\text{SST} = 9.328(0.729 - (\text{SIC}/100)^3) - 1.8 \, , \tag{1}$$

- SIC was set to zero if SST > 4.97 °C.
- Where -1.8 °C < SST < 4.97 °C we adjusted SIC as follows:

$$SIC = 100(0.729 - (SST + 1.8)/9.328)^{\frac{1}{3}} \, , \tag{2}$$


The analysis presented below includes comparisons between the two PGW simulations and the control simulation (Fig. 1). Following Noël et al. (2014), we allotted 5 years of spin-up time for each model simulation to allow the MAR snowpack model to adjust to the altered boundary conditions. In PGW1, we adjusted the boundary forcing fields of temperature, specific humidity, SST, and SIC to reflect the long-term preindustrial conditions using the procedures detailed above. Thus, by
comparing PGW1 to the control simulation, we quantify the thermodynamic contribution to recent GrIS surface mass loss. For PGW2, we adjusted SST and SIC to reflect preindustrial conditions, while leaving the temperature and humidity fields unaltered. In doing so, we quantify the portion of recent surface mass loss that is due to changes in local sea-surface conditions alone.

The design of PGW2 also allows us to test the theory of Stroeve *et al*. (2017) that low spring SIC in the seas surrounding Greenland preconditions the GrIS for melt in early melt onset years. Following Stroeve *et al*. (2017), we define melt onset as the first instance of five or more consecutive days of melt. The date of freeze onset is then defined as the first day following the last instance of five or more consecutive days of melt and melt season length as the number of days spanning the two dates. We calculated all measures of the melt season at each MAR grid pixel, then tested for significant differences between the PGW
simulations and the control using a paired Wilcoxon signed-rank test (Wilcoxon, 1945) with a predetermined significance level of $\alpha = 0.05$ (i.e., 95 % confidence level).

## 3. Results

### 3.1. Thermodynamic Contribution to GrIS SMB Change

Figure 4 presents a comparison of the cumulative SMB anomaly between the control run and each of the PGW simulations.
The control run (Fig. 4, gray line) shows a cumulative SMB anomaly of -1852 Gt over the study period of 2000 to 2019, congruent with other estimates (IMBIE, 2020). This decline in the SMB corresponds to approximately 5 mm of global sea





level rise. A gradual shift to a negative cumulative SMB occurs around 2005, coinciding with the transition to a more persistently negative NAO and rise in Greenland blocking frequency (Hanna et al., 2015; Hofer et al., 2017). The first instance of pronounced mass loss is evident as a sharp decrease in 2007—a year of unprecedented surface melt up to that point in the
satellite record (Mote, 2007). Instances of marked mass loss are frequently evident in the years that follow; however, the exceptional melt years of 2012 (Hanna et al., 2014; Nghiem et al., 2012) and 2019 (Cullather et al., 2020; Tedesco and Fettweis, 2020) are readily apparent as precipitous drops in the control time series. We examine these two exceptional melt years in more detail in section 3.4.

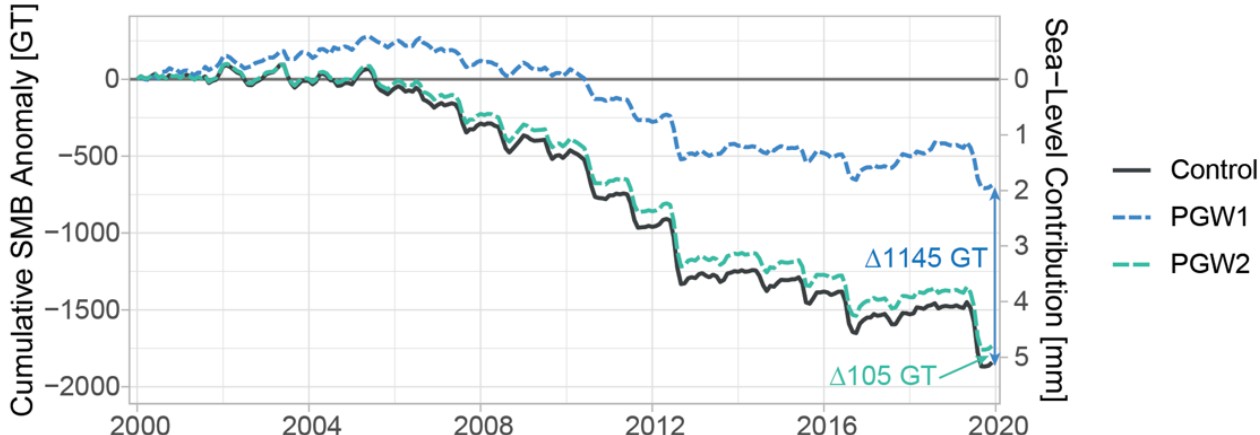

**Figure 4. Temporal evolution of the GrIS SMB under contrasting thermodynamic background conditions.** Shown are the cumulative SMB anomaly time series for the control (gray), PGW1 (blue dashed), and PGW2 (green dashed) simulations. Anomalies calculated with respect to the 1980-1989 reference period. Left axis shows cumulative SMB anomaly; right axis shows the equivalent sea-level contribution. Annotations detail the difference in the final cumulative SMB between each of the PGW simulations and the control.

Comparing the control with PGW1 (Fig. 4, blue dashed line) clearly highlights the substantial thermodynamic contribution to the recent change in GrIS SMB. A difference in cumulative SMB between the two simulations of 1145 Gt amounts to a 62% reduction in surface mass loss in PGW1 relative to the control. Under the preindustrial thermodynamic setting of PGW1, the GrIS maintains a positive SMB anomaly through 2009, and while the dynamical forcing of the ice sheet is still evident in the negative anomalies that occur during individual summers, the mass loss for each melt season is much more subdued relative
to the control. This holds true for the exceptional melt years of 2012 and 2019; however, while the magnitude of mass loss is greater when the anthropogenic warming signal is included, the relative contribution of those individual melt seasons to the total SMB change over the 20-year period is greater for PGW1—In a preindustrial climate, 2012 and 2019 each account for ~250 Gt of mass loss, which – combined – is approximately 2/3 of the total mass loss in PGW1 (Fig. 4). Furthermore, while the rate of mass loss is much reduced from 2013 to 2018 in the control, this period undergoes a slight surface mass gain in
PGW1.



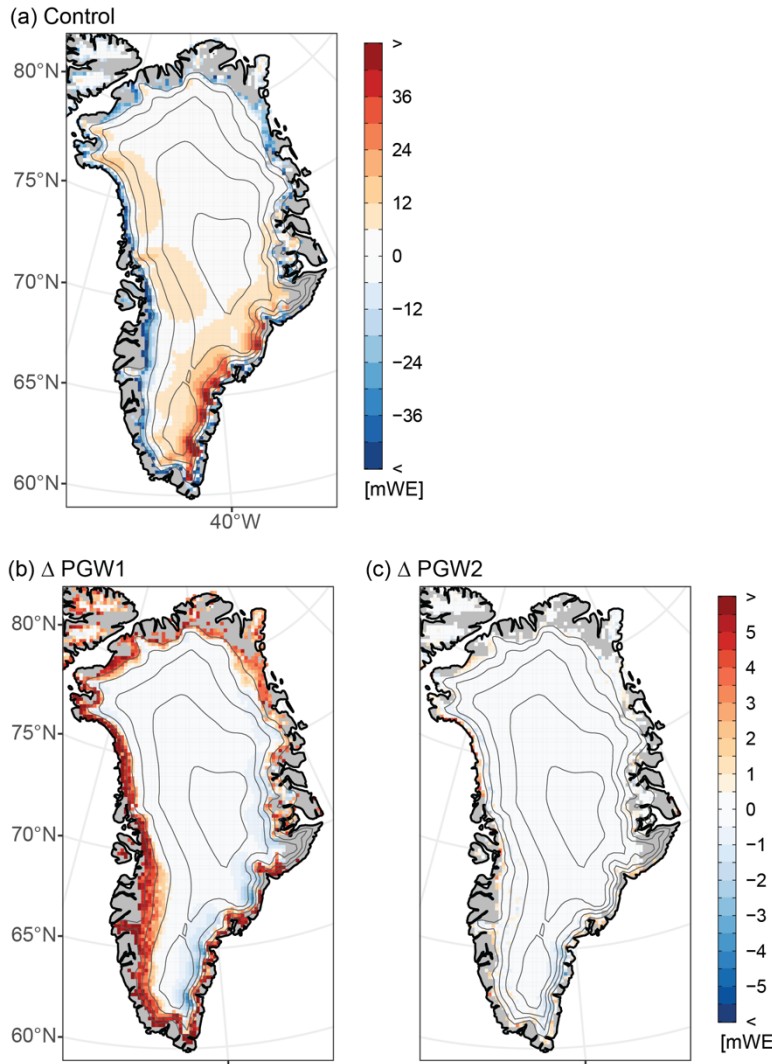

**Figure 5. Spatial distribution of GrIS SMB change under contrasting thermodynamic background conditions.** (a) The cumulative SMB anomaly over the full study period of 2000–2019 as represented by the control simulation. (b) PGW1 cumulative SMB minus the control. (c) PGW2 cumulative SMB minus the control. Contour interval: 500 m. Range: 1000–3000 m.


The cumulative SMB anomaly in PGW2 (Fig. 4, green dashed line) is 105 Gt greater than that for the control. This relatively small difference indicates that there has been minimal influence by changes in local SST and SIC over the study period—a result that is consistent with previous modeling studies which showed low GrIS sensitivity when applying arbitrary perturbations to local sea-surface conditions (Hanna et al., 2009, 2014; Noël et al., 2014). This agreement across varying methodological approaches adds confidence that changes occurring more widely throughout the Arctic and sub-Arctic dominate the thermodynamic contribution to GrIS mass loss.






Figure 5 provides a spatial representation of the comparisons made in Fig. 4. The map of cumulative SMB over the study period as modeled by the control (Fig 5a) shows a band of negative SMB along the perimeter of the ice sheet that clearly

demarcates the ablation zone, where annual surface mass loss exceeds accumulation. The greatest accumulation occurs along the southeast coast of Greenland and is a product of orographic enhancement of precipitation associated with lee-side cyclones that form in westerly flow over southern Greenland (Bromwich et al., 1998; Rogers et al., 2004; Schuenemann et al., 2009). Other areas of notable SMB gains include west and northwest Greenland. Snow accumulation in these areas is fueled by bouts of intense water vapor transport through the Davis Strait that have increased in frequency in recent decades (Mattingly et al.,

2016, 2018).

Relative to the control, PGW1 yields a greater cumulative SMB in a band that stretches around the perimeter of the ice sheet, exceeding 2000 m elevation in some locations in southwest Greenland (Fig 5b). This positive anomaly with respect to the control is a consequence of decreased meltwater runoff in the preindustrial setting (Supplementary Fig. 2a). At higher

elevations over much of eastern Greenland and to a lesser extent over the northwest ice sheet, a reduction in snowfall in the cooler and dryer atmosphere of PGW1 results in a lower SMB compared to the control (Fig 5b, Supplementary Fig. 2b).

Figure 6 shows the average seasonal progression of the principal SMB components for each model simulation. The greatest differences in surface runoff between PGW1 and the control are centered on the peak of the melt season in mid-to-late July

(Fig. 6a). The gray shading in Fig. 6 depicts the 1 std. dev. range about the mean that was simulated for each variable across the 20-year control run to provide context regarding magnitude of the differences between the experiment and the control. This reveals that runoff during the peak of the melt season in PGW1 was nearly 1 std. dev. below what has been typical since the turn of the century. The relative mass loss over high elevations evident in Fig 5b is driven by a reduction in snowfall throughout the cool season; however this impact on snow accumulation is most apparent in fall and early winter when the greatest change

in background conditions have occurred under Arctic amplification (Fig. 2, 7b) (Serreze and Barry, 2011). In contrast with the rest of the year, there is a slight increase in summer snowfall in PGW1 that coincides with a reduction in rainfall (Fig. 6c), consistent with greater partitioning toward frozen precipitation under the cooler preindustrial setting.

The differences between PGW2 and the control show a similar pattern as observed for PGW1; however, they are comparatively

minimal in both magnitude and scale (Fig 5c). The change in sea-surface conditions in PGW2 reduces meltwater runoff resulting in higher SMB (Fig 5c, Supplementary Fig. 1d). Unlike PGW1, this response is largely confined to grid cells along the terminus of the ice sheet. The isolated impact of sea-surface conditions on snowfall is most evident along the southeast margin of the ice sheet and above ~1000 m in northwest Greenland. In contrast with PGW1, in which runoff was diminished throughout the entire melt season, the impact of sea-surface conditions alone on surface melt emerges later in the melt season



(Fig. 6a), likely reflecting the stronger coupling between ocean and atmosphere as the thermal gradient between them increases

into the fall (Screen, 2017).

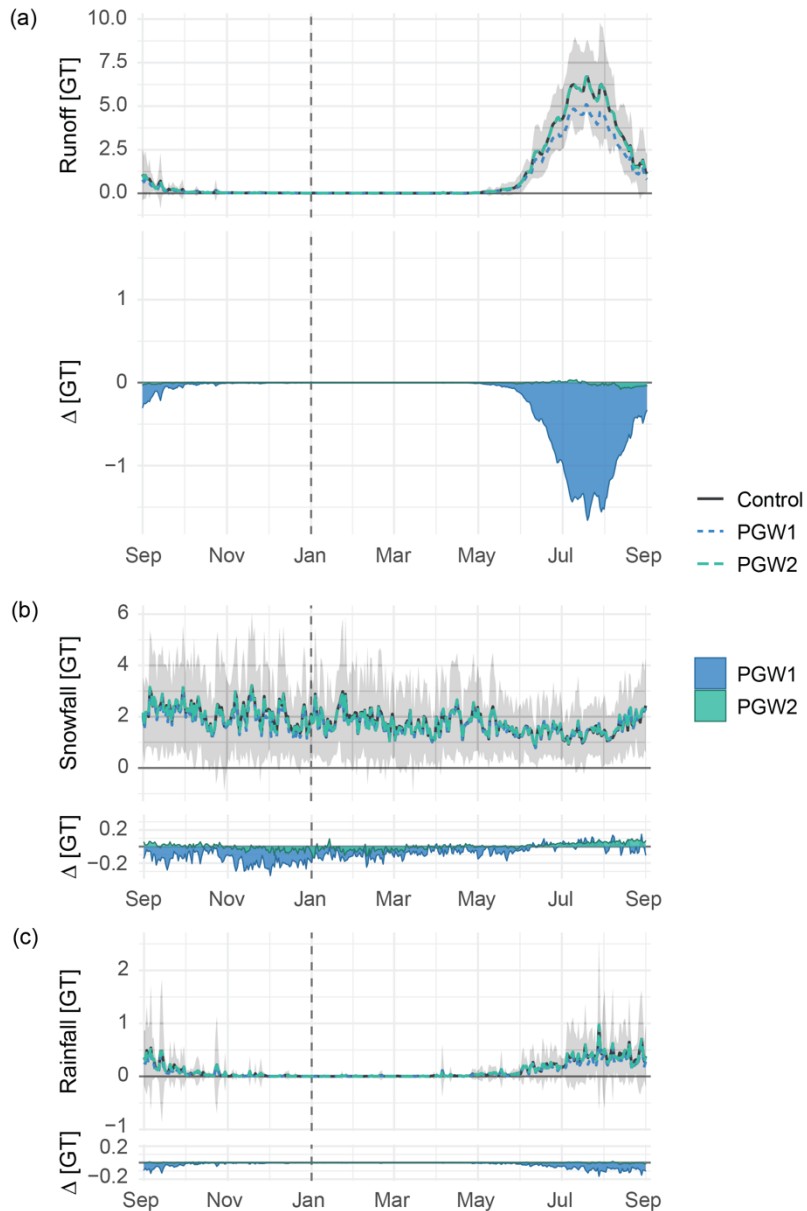

**Figure 6. Seasonal evolution of GrIS SMB under contrasting thermodynamic background conditions.** Panels depict the seasonal
progression of three principal SEB components: (a) Surface runoff, (b) snowfall, (c) rainfall. Top portion of each panel shows 2000–2019

long-term daily mean totals of each SMB component throughout the melt season for the control (gray), PGW1 (blue dashed), and PGW2
(green dashed) simulations. Time series represent the spatially integrated sum of a given variable over the entire ice mask. Gray shading
shows the 1σ range about the mean for the control simulation. Bottom portion shows the difference between each PGW simulation (PGW1,
blue; PGW2, green) and the control (Δ = PGW − Control). The scale of the y-axis on the bottom portion is kept constant across all panels to
facilitate comparison between SMB terms.






**Figure 7. The SEB of the GrIS during the melt season under contrasting thermodynamic background conditions.** (a–d) Top portion of each panel shows the 2000–2019 long-term daily mean values of each SEB component throughout the melt season for the control (gray), PGW1 (blue dashed), and PGW2 (green dashed) simulations. Time series represent the spatial average taken over the entire ice mask for a given variable. Gray shading shows the 1σ range about the mean for the control simulation. Bottom portion shows the difference between





each PGW simulation (PGW1, blue; PGW2, green) and the control (Δ = PGW−Control). The scale of the y-axis on the bottom portion is kept constant across all panels to facilitate comparison between SEB terms. (e–f) Maps depicting the difference between the 2000–2019, May–Sep long-term mean of each SEB component between each PGW simulations (PGW1, left; PGW2, right) and the control. SEB components are organized by row: (a, e) downward shortwave radiation (SWD); (b, f) downward longwave radiation (LWD); (c, g) sensible heat flux (SHF); (d, h) latent heat flux (LHF). Contour interval: 500 m. Range: 1000–3000 m.


While a decrease in snowfall relative to the control in the PGW simulations partially compensates for the relative mass gains at lower elevations, it is clear that the reduction in meltwater runoff is the primary determinant of the differences in cumulative SMB observed in Fig. 4. Thus, the influence of recent thermodynamic change on GrIS SMB has been consequential during the melt season. Recognizing this, the next section focuses on the extended melt season to better understand the mechanisms

by which thermodynamic change has dictated GrIS surface runoff.

### 3.2. Thermodynamic Drivers of GrIS Surface Runoff

Figure 7 contrasts the SEB of the GrIS between the control and PGW simulations. The preindustrial thermodynamic state of PGW1 is associated with an increase in downward shortwave radiation (SWD) (Fig. 7a, e) and a decrease in downward longwave radiation (LWD) (Fig. 7b, f) throughout the melt season. For both variables, the differences between the control and

PGW1 are greatest over the northern ice sheet. This is consistent with the thermodynamic signature in the free atmosphere where the differences in both temperature and specific humidity at 600 hPa are maximized over northern Greenland (Fig. S3). The time series in Fig. 7c and d indicate that the turbulent fluxes are generally diminished in PGW1 relative to the control. The magnitude of these differences is far less than what is observed for the radiative terms; however, this is partly attributable to a spatially heterogeneous response. Differences in the turbulent heat fluxes are focused along the outer margins of the ice

sheet, where lower elevations display a decrease in both sensible (SHF) and latent heat flux (LHF) in PGW1 that is mirrored by differences of the opposite sign over higher elevations (Fig. 7g, h).

The juxtaposition in the response of the turbulent fluxes in PGW1 appears to arise from opposing direct and indirect responses to the change in background conditions. Along the ice sheet margins, a decrease in both SHF and LHF is consistent with a

direct reduction in the flux of heat and moisture to the surface of the ice sheet in a colder, drier preindustrial atmosphere. Conversely, above normal turbulent fluxes over higher elevations follows indirectly from changes in the near-surface wind field. Figure 8a shows the long-term mean May–Sep 10 m winds for the control period, clearly illustrating the persistent katabatic wind signature over Greenland. Lower water vapor content in PGW1 (Fig. 2) reduces the longwave emissivity of the atmosphere, which would act to lower surface temperatures, increase the near-surface potential temperature deficit, and thereby

strengthen the katabatic winds over the upper portion of the steep margins of the GrIS (Fig. 8b) (van den Broeke et al., 2009b; Gorter et al., 2014). Stronger katabatic winds then increase turbulent heat flux by mixing relatively warm air through the stable boundary layer to the surface (Fig. 7g, h).



While the differences in the radiative terms considerably outweigh that of the turbulent fluxes in PGW1, this is not the case
for PGW2. Although the minor differences in SWD and LWD are more widespread (Fig. 7e, f), the magnitude of the impact
of sea-surface conditions on turbulent heat flux is greater in some locations along the ice sheet margins, particularly as is
evident in the reduction in SHF along the northern and central portions of the western ablation zone (Fig. 7g). This appears to
be primarily a consequence of the indirect katabatic wind adjustment. The lower SST and higher SIC in PGW2 reduces the
horizontal temperature gradient between the ice sheet and surrounding seas which, as has been documented in previous work
(Noël et al., 2014), causes a weakening of the katabatic wind along the ice sheet margins (Fig. 8c). The decrease in the near-
surface wind field would reduce turbulent mixing, and thus SHF, to the surface, while also causing a reduction in evaporation
/ sublimation, resulting in an increase in LHF relative to the control.

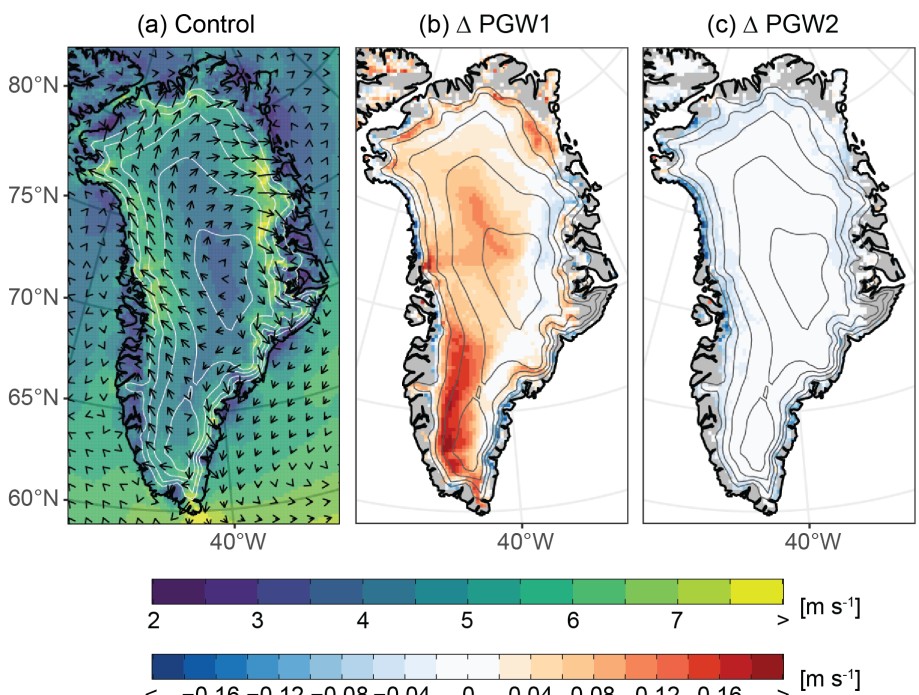

**Figure 8: Melt season katabatic wind field under contrasting thermodynamic background conditions.** (a) 2000–2019 long-term mean
May–Sep 10 m wind speed (shading) and direction (vectors). (b) Difference in 10 m wind speed between PGW1 and the control. (c)
Difference in the 10 m wind speed between PGW2 and the control. Δ = PGW − Control. Contour interval: 500 m. Range: 1000–3000 m.

The consistent and widespread reduction in LWD that is visible for PGW1 (Fig. 7b and f) is not surprising given the drier
atmospheric conditions that prevailed during the preindustrial period. The specific humidity perturbation signal that was
applied to PGW1 (see Fig. 2) is manifest in the integrated water vapor (IWV) over the ice sheet (Fig. 9a, e). The spatial
distribution of the IWV anomalies with respect to the control closely resemble that for LWD (c.f. Fig. 9e and 7f)—highlighting
the effectiveness of water vapor as a greenhouse gas and nicely illustrating the cause of the longwave radiative response. It is
important to note that the LWD anomalies in Fig. 7b and f do not provide a complete picture of the end impact on the SEB, as





any resulting change in the temperature of the ice sheet's surface would be offset to some degree by a change in emitted longwave radiation in accordance with the Planck feedback. Indeed, this can be seen in Fig. S4, which depicts a weaker and
less uniform response across the ice sheet when considering the difference in net longwave radiation between PGW1 and the control; however, it remains the case that the preindustrial setting of PGW1 produces reductions in net longwave that are most evident over the northern GrIS.

The consequence of this water vapor feedback can be seen in the ice-sheet-wide drop in surface temperature simulated by
PGW1. Here, the differences in the seasonality, magnitude, and spatial distribution of the maximum (Tmax) and minimum (Tmin) daily temperature response reflects seasonal and diurnal variations in the relative importance of longwave radiative effects, as well as geographical variations in surface albedo (Lenaerts et al., 2019; Wang et al., 2018, 2019). The magnitude of the Tmax and Tmin anomalies in PGW1 relative to the control both increase from spring into fall (Fig. 9b, c). This seasonal pattern is consistent with stronger Arctic amplification, and thus a greater water vapor feedback, in the fall than in spring as
pan-Arctic reductions in SIC in a warmer climate allow for increased heat flux from the ocean to the comparatively cool fall atmosphere (Chung et al., 2021). Additionally, there is a decline in incoming shortwave radiation as the solar declination decreases into winter, which elevates the relative contribution of longwave radiative effects to the SEB. There is a distinct north-south gradient in the maximum daily surface air temperature (Tmax) response (Fig. 9f). The weaker Tmax differences over southern Greenland in PGW1 are a consequence of both a higher sun angle at lower latitudes and the lower surface albedo
of the southern ice sheet, both of which decrease the relative longwave contribution to the SEB. The impact of surface albedo is particularly evident in the weak Tmax response over the southwest ablation zone (Fig. 9f). At night, LWD constitutes the sole radiative input to the SEB. Consequently, the Tmin response is notably greater than Tmax and it more closely resembles that of IWV (Fig. 9e) and LWD (Fig. 7f).

PGW1 exhibits a band of higher surface albedo throughout the melt season that runs along the perimeter of the GrIS (Fig. 9d, h) and closely aligns with areas where IWV (Fig. 9e), Tmin, (Fig. 9g), SHF (Fig. 7g), and surface runoff (Supplementary Fig. 1a) are reduced in PGW1 relative to the control. Thus, the longwave radiative response to reduced water vapor content combined with diminished SHF in a cooler atmosphere appear to be critical factors contributing to lower surface runoff under the preindustrial setting of PGW1. The reduction in water vapor decreases LWD, which allows for lower Tmin. These changes
would reduce runoff directly, by increasing the portion of meltwater that is refrozen within the snowpack, and indirectly, by diminishing the ice-albedo feedback, thereby impacting the shortwave components of the SEB. The interdependence between SEB components is effectively illustrated by the differences in net shortwave radiation between PGW1 and the control (Fig. S4)—the magnitude of the differences in net shortwave radiation clearly exceeds that for SWD along the perimeter of the ice sheet, emphasizing the importance of the ice albedo feedback to the thermodynamic contribution to GrIS surface mass loss.
That the strongest signal in these variables is located over the northern Greenland and aligned with some of the largest increases





in surface albedo supports previous work demonstrating the importance of this longwave radiative mechanism to runoff from the northern GrIS (Noël et al., 2014).

**Figure 9. Thermodynamic mechanisms of GrIS SMB change.** (a–d) Top portion of each panel shows the 2000–2019 long-term daily mean values of each variable throughout the melt season for the control (gray), PGW1 (blue dashed), and PGW2 (green dashed) simulations. Gray shading shows the 1σ range about the mean for the control simulation. Time series represent the spatial average taken over the entire



ice mask for a given variable. Bottom portion shows the difference between each PGW simulation (PGW1, blue; PGW2, green) and the control (Δ = PGW −Control). (e–f) Maps depicting the difference between the 2000–2019, May–Sep long-term mean of each variable between the PGW simulations (PGW1, left; PGW2, right) and the control. Variables are organized by row: (a, e) integrated water vapor (IWV); (b, f) daily maximum surface air temperature (Tmax); (c, g) daily minimum surface air temperature (Tmin); (d, h) surface albedo (ALB). Contour interval: 500 m. Range: 1000–3000 m.

Focusing on PGW2, it is evident that sea-surface conditions alone exert minimal influence on the ice sheet. There is no clear pattern of influence on IWV or near surface air temperature (Fig. 9e–g), and an examination of temperature and humidity at

600 hPa shows no evidence of any appreciable influence on these variables in the free atmosphere (Fig. S3). There is, however, an increase in surface albedo along the western and northern margins of the ice sheet in PGW2 relative to the control that occurs late in the melt season (Fig. 9d, h) and appears to be the product of the collocated reduction in SHF (Fig. 7g).

### 3.3. Thermodynamic Change and GrIS Melt Timing

Consistent with previous studies (Hanna et al., 2009, 2014; Noël et al., 2014), the above results suggest that local marine

influence on GrIS melt is limited to the outermost margins of the ice sheet. Furthermore, these results demonstrate that the influence of local sea-surface conditions is an order of magnitude less than what is observed for the full thermodynamic forcing of PGW1 (Fig. 4). This, combined with the lack of a IWV or surface air temperature signal over the ice sheet in PGW2 (Fig. 9),  appears to contradict the results in Stroeve *et al*. (2017) that early sea ice loss in Baffin Bay may have an appreciable impact on meltwater production in extreme melt years by preconditioning the ice sheet for early melt onset via downward

longwave radiative forcing.

To examine this hypothesis more directly, Fig. 10 presents the results of a paired, signed-rank test comparing differences in median GrIS melt timing between the control and each of the PGW simulations. At lower elevations, where melt occurs consistently on an interannual basis, melt onset during the 2000–2019 study period typically occurs between early-May and

mid-June (Fig. 10a) while freeze onset occurs from early-August through September (Fig. 10b). Later melt onset and earlier freeze onset is evident over higher elevations; however, melt in these regions is typically short-lived (Fig. 10c) and infrequent. Accordingly, and as mentioned in the experimental design, the comparisons of melt timing between the PGW simulations and the control are limited to lower elevation locations with a sufficient sample of years experiencing melt.

Relative to the control, the median date of melt onset in PGW1 occurs, on average, ~2.5 days later across those regions of the GrIS that consistently experience melt, while in the upper quartile, grid cells experienced delays in median melt onset of ≥4 days (Fig. 10d). The thermodynamic impact on the close of the melt season was even greater—the median date of freeze onset advanced, on average, by ~3.7 days and freeze onset in the upper quartile of grid cells shifted to ≥5.5 days earlier in the fall (Fig. 10e). Combined, these changes shortened the median melt season duration by an average of ~6.7 days, while melt duration

in the upper quartile of grid cells shortened by ≥9 days. For all melt timing metrics, the differences between the PGW1 and the control that were deemed statistically significant at the 95% confidence level are widespread across the examined grid cells



### 3.4. The Exceptional Melt Years of 2012 and 2019

Embedded in the long-term GrIS SMB decline (Fig. 4), 2012 and 2019 stand out as exceptional years of surface mass loss. According to the control simulation, there was a cumulative SMB anomaly of -364 Gt during the 2011–2012 hydrological year
(Fig. 11a). The melt season of 2012 was characterized by recurrent episodes of intense surface runoff (Fig. 11b), spurred by anomalous atmospheric forcing of the ice sheet (Hanna et al., 2014). During this stretch of melt events, pronounced atmospheric ridging over Greenland promoted southerly advection of warm, moist air from low latitudes to the western ice sheet (Hermann et al., 2020; Neff et al., 2014), generating strong turbulent heat fluxes that drove high-volume meltwater production over the western ablation zone (Cullather et al., 2020; Fausto et al., 2016b). Adiabatic cooling of remotely-sourced
moist air that ascended the western slope of the GrIS on July 12 prompted the formation of low-level, liquid clouds that supplied the requisite longwave radiative forcing for widespread melt over high elevations (Bennartz et al., 2013; Neff et al., 2014), generating a single day melt extent that covered over 98% of the ice sheet's surface (Nghiem et al., 2012).

The cumulative SMB anomaly over the 2018–2019 hydrological year totaled -376 Gt (Fig. 11e). The melt season of 2019 was
heavily influenced by a blocking anticyclone, with origins in the European heatwave of the same year (Cullather et al., 2020), that produced tremendous surface runoff during a melt event centered around July 31 (Fig. 11f). The air mass, which was transported west from Europe, was warmer and drier in comparison with that which was responsible for the mid-July, 2012 melt event and, consequently, did not produce the same low-level cloud cover that was instrumental to melt of the accumulation zone in 2012 (Tedesco and Fettweis, 2020). Consequently, while the total surface mass loss in 2019 was comparable to that of
2012, observed melt was not as extensive in 2019, reaching a maximum coverage of ~73% of the ice sheet's surface on July 31 (Tedesco and Fettweis, 2020).

For both years, the portion of observed GrIS surface mass loss that is attributable to changes in the local background thermodynamic environment was far less than the average for the study period: whereas the total mass loss over the entire
2000–2019 study period was ~62% less in PGW1 relative to the control, the reduction in mass loss was a relatively modest 30% and 25% in 2012 and 2019, respectively (Fig. 11a, e). This suggests that the relative importance of a changing background state under global climate change may be minimized during periods of strong synoptic-scale atmospheric forcing. In other words, the record melt observed during those two summers is more a consequence of exceptional atmospheric circulation patterns than it is a direct consequence of the long-term warming trend; however, it is important to note that these exceptional
circulation patterns and the long-term temperature trend may not be independent, as some studies have suggested more persistent circulation regimes under global warming (Coumou et al., 2018; Preece et al., 2023b; Screen, 2013). Indeed, this disparity is also evident over synoptic timescales—the periods of strong dynamical forcing of the GrIS, marked by the red, vertical bars in Fig. 11, correspond to local minima in the differences in daily-mean near-surface air temperature between





PGW1 and the control. The production of meltwater and consequent surface runoff during high-volume melt events is largely
driven by turbulent heat fluxes (Fausto et al., 2016a, b). It follows that the longwave radiative effects of the water vapor
feedback that are strongly dictated by changes in the thermodynamic environment assume a lesser role during these periods of
intense melt. Consistent with the results for the full study period, the minimal difference between PGW2 and the control
suggests no appreciable contribution by the observed change in local sea-surface conditions to runoff production during these
exceptional melt years.

**Figure 11. Thermodynamic contribution to surface mass loss during years of exceptional GrIS melt.** Panels show (a, e) the cumulative
SMB anomaly spanning the Sep–Aug hydrological year alongside (b, f) total daily meltwater runoff, (c, g) mean daily near-surface air





temperature anomaly, and (d, h) mean daily integrated water vapor anomaly during the exceptional melt years of (a–d) 2012 and (e–h) 2019.
In all panels, time series are presented for the control (gray), PGW1 (blue dashed), and PGW2 (green dashed) simulations. Bottom portion
of b–d and f–h shows the difference between each PGW simulation (PGW1, blue; PGW2, green) and the control (Δ = PGW − Control). Red
vertical shading highlights periods of strong synoptic-scale forcing. Cumulative anomalies in (a, e) calculated with respect to the 1980–1989
reference period. Anomalies in (b-d, f-h) calculated with respect to the entire 2000–2019 study period and represent the spatial average taken
over the entire ice mask for a given variable.

Meltwater that exits the ice sheet as runoff is primarily sourced from the ablation zone and is therefore controlled by processes,

such as turbulent heat flux and incoming solar radiation (due to the low albedo), that exert a strong influence along the margins
of the GrIS. As was highlighted above, melt in the high-elevation accumulation zone (with high albedo) is more dependent on
longwave radiative effects and presence of clouds. Figures 11d and h show that atmospheric water vapor content is consistently
reduced throughout both 2012 and 2019 in PGW1 relative to the control, particularly during periods outside the highlighted
instances of strong synoptic-scale atmospheric forcing. Thus, the influence of thermodynamic change during these years of

extreme mass loss may be more visible in the frequency of melt over the accumulation zone than for total meltwater runoff.

To investigate this possibility, Fig. 12 presents a comparison of the number of days that underwent surface melt during the

melt seasons of 2012 and 2019 in each of the model simulations. Focusing on the control (Fig. 12a, c), melt frequencies across

the ice sheet were generally greater in 2012 than 2019. The difference between the two years is quite evident over the southern

portion of the ice sheet, where locations above 2500 m in elevation recorded over 40 days of melt in 2012 (Fig. 12a). Melt was

also more frequent above ~1500 m over the northern GrIS in 2012, but 2019 underwent more frequent melt at lower elevations

of the most northern margin of the ice sheet. These results align with previous work demonstrating greater runoff from the

northern drainages of the GrIS but lower total melt extent in 2019 relative to 2012 (Cullather et al., 2020; Tedesco and Fettweis,

2020).


Broadly across the GrIS, the difference between PGW1 and the control show that the thermodynamic contribution to melt

frequency was greater in 2019 than in 2012 (Fig. 12b, d). Unlike 2019, intense water vapor transport accompanied the extensive

melt events of 2012 (Hermann et al., 2020; Neff et al., 2014; Tedesco and Fettweis, 2020). Thus, the longwave radiative forcing

necessary for melt over the high-albedo accumulation zone was provided by anomalous atmospheric moisture supplied by the

large-scale circulation, which likely resulted in less sensitivity to changes in local thermodynamic conditions. In both years,

the greatest differences in melt frequency between PGW1 and the control are located just above the ablation zone, where

perennial snow cover raises the albedo of the ice sheet, making it more susceptible to the longwave radiative effects that

accompany the change in background conditions (Fig. 12b, d). The spatial distribution of the differences in melt frequency in

PGW1 also highlights the thermodynamic contribution to the maximum elevation of GrIS melt extent. This is most evident

when considering the decline in melt frequency over high elevations in the context of the total number of melt days simulated

by the control. While the reduction in melt frequency of 1 to 5 days at elevations above ~2000 m in northern and central

Greenland is low compared to other regions of the ice sheet, it is comparable to the total observed number of melt days

simulated by the control (c.f. Fig. 12a, c and b, d), demonstrating that melt over much of the high accumulation zone would not have occurred if not for recent climate warming. Contrary to PGW1, the changes in the number of PGW2 melt days relative

to the control are minimal and generally do not exhibit a coherent spatial signal (Fig. 12b, d); however, there is some indication of a decline in 2019 melt frequency over the western slope of the southern ice sheet that is opposed by an increase in melt frequency above 2000 m (Fig. 12d).

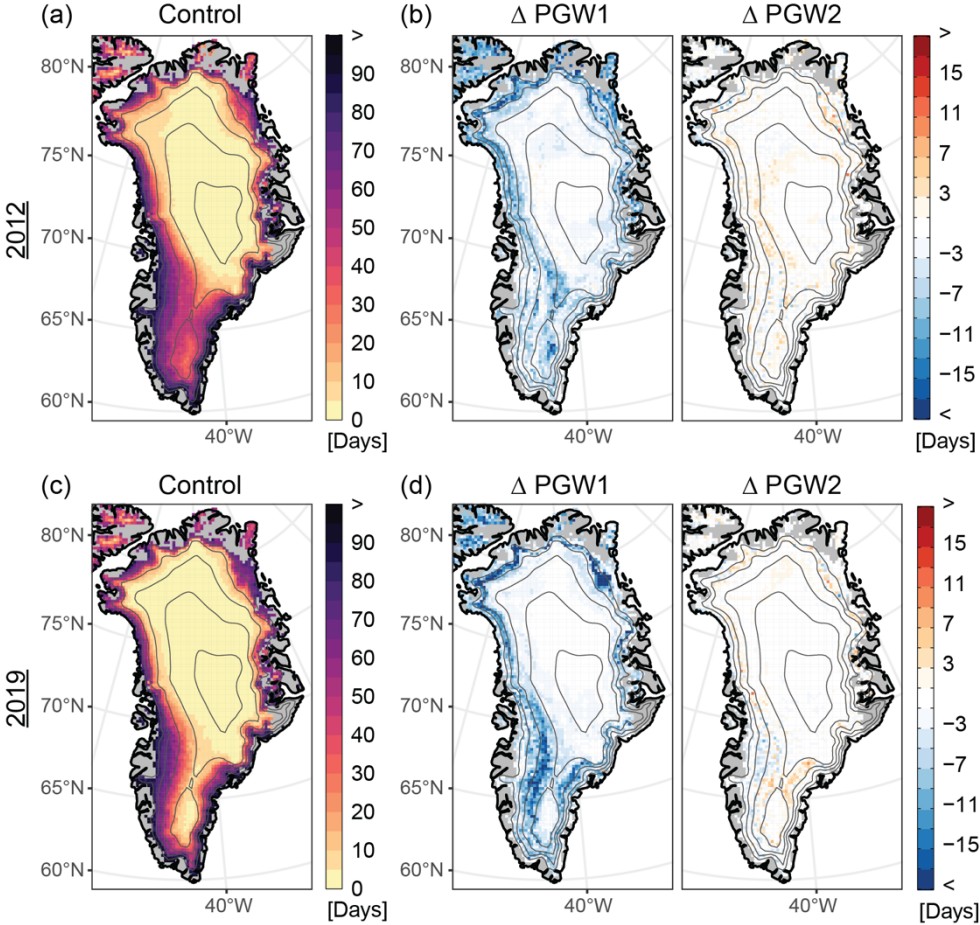

**Figure 12. Comparison of GrIS melt frequency between PGW simulations during years of exceptional GrIS surface mass loss.** (a, c) The number of days in which the control simulation indicated ≥5 mm melt in a given grid cell and (b, d) the difference between the PGW
simulations and the control (PGW1, left; PGW2, right) for the exceptional melt years of (a, b) 2012 and (c, d) 2019. Δ = PGW − Control Contour interval: 500 m. Range:1000–3000 m

## 4. Discussion and Conclusions

Much of the work examining the recent, pronounced increase in GrIS meltwater runoff has rightfully focused on the role of atmospheric dynamics in facilitating this change (Bevis et al., 2019; Fettweis et al., 2013; Hanna et al., 2015, 2016, 2018b,
2022; Hofer et al., 2017). While some have presented evidence of a relationship between global climate change and the shift in summer atmospheric circulation that has promoted melt of the GrIS (Liu et al., 2016; Preece et al., 2023b; Screen, 2013), a



conclusive link remains a subject of investigation. In contrast, the accelerated rate of warming in the Arctic represents a robust climate change signal that has undoubtedly contributed to recent GrIS SMB trends (Boers and Rypdal, 2021; Hanna et al., 2008). This work represents, to our knowledge, the first systematic attempt to quantify the contribution of the local change in
background thermodynamic conditions to recent GrIS surface mass loss.

Our results indicate that had recent atmospheric dynamical forcing occurred in a preindustrial climate, surface mass loss from the GrIS would have been reduced by over 62% (Fig. 4), highlighting the substantial contribution of external climate forcing. The mechanisms by which local thermodynamic background conditions contribute to SMB change appear to be dominated by
longwave radiative effects stemming from the water vapor feedback. The amplified rate of warming in the Arctic has augmented surface runoff by promoting an increase in atmospheric moisture content and associated downwelling longwave radiation (Fig.9a, e and 7b, f), which disproportionately increases daily minimum temperatures (Fig. 9c, g). Combined with increased SHF along the margins of the GrIS (Fig. 7c, g), these changes have reduced the surface albedo (Fig. 9d, h), further increasing meltwater runoff (Fig. 6a, S4). These results are consistent with the recent findings of Noël *et al.* (2019), who show
that an increase in downwelling longwave radiation has caused a disproportionate increase in surface runoff from the northern drainages of the GrIS by efficiently promoting melt and expanding the ablation zone in this region of high albedo, as well as by increasing daily minimum temperatures, which reduces meltwater refreeze within the firn layer. While the authors point to the advection of moisture-rich air to the northern ice sheet by anomalously anticyclonic summer circulation over Greenland, the results of this analysis suggest that the increase in background temperature constitutes an important contribution to this
mechanism on its own.

The 62% reduction in surface mass loss under preindustrial conditions presented here does not imply that atmospheric circulation is only responsible for 38% of the observed impact on GrIS SMB, as the individual contributions of atmospheric dynamics and thermodynamics should sum to the *total* change in SMB relative to what it would have been if neither an
increased frequency of Greenland blocking nor anthropogenic warming had occurred. Given that the GrIS maintained a positive cumulative SMB anomaly through 2009 under the preindustrial thermodynamic background conditions imposed in PGW1, it is possible that the cumulative anomaly may have remained positive through the end of the study period if not for the increased frequency of high-amplitude circulation patterns. Thus, relative to this hypothetical preindustrial climate with more typical atmospheric circulation, the total change in SMB would be greater than the magnitude of the negative anomalies
presented in Fig. 4 and the contribution of atmospheric circulation to this total change would exceed 38%. Indeed, using an earth system model to nudge the wind field toward observed conditions while maintaining constant external forcing, Topál et al. (2022) showed that changes in atmospheric circulation explained 56% of the increase in surface air temperature over Greenland from 1990 to 2012. Likewise, using historical data and a circulation analogue technique, Fettweis et al. (2013) found that the shift in summer circulation explained ~70% of the 1993–2012 warming at 700 hPa over Greenland. Speaking
more directly to mass loss, Delhasse et al. (2018) compared output from MAR forced by perturbed reanalysis data from the



recent period of increased Greenland blocking against simulations forced by output from GCMs which have collectively failed to capture this change in circulation. Their results suggest that, if the recent anomalous circulation persists into the future, the GrIS will undergo more than twice the surface mass loss that is currently projected by GCMs. Thus, understanding this circulation change and why it is not represented in climate models must be a top priority for accurate projections of GrIS mass
loss.

The contribution of local thermodynamic background conditions to total surface mass loss during the exceptional melt years of 2012 and 2019 was less than half that which was observed for the entire 2000–2019 study period (c.f. Fig. 4 and 11a, e), demonstrating that the relative contribution by the local thermodynamic state is reduced during periods of strong large-scale
atmospheric forcing. This is also evident over synoptic timescales, where the difference in surface air temperature between PGW1 and the control is minimized on days of exceptionally high-volume surface runoff; rather, the greatest differences in surface temperature appear to emerge during periods encompassing temporal minima in air temperature (Fig. 11). This pattern of influence likely reflects the increased contribution of remotely sourced heat and moisture during strong large-scale forcing— reducing the relative importance of the background thermodynamic state—as well as the efficacy of the longwave radiative
effects that typify the response to changes in the thermodynamic background state in regulating minimum temperatures. In other words, recent local thermodynamic change around Greenland appears to have promoted GrIS surface runoff by raising the floor of the temperature distribution more so than by exacerbating warm extremes.

It is important to note that the same large-scale atmospheric conditions that typify our control period and have encouraged
GrIS mass loss have also fostered below-normal sea ice in the region (Ballinger et al., 2018; Ogi and Wallace, 2007; Stroeve et al., 2017). Thus, the 1880–1899 sea ice climatology that we prescribe here may often exceed the SIC that would have occurred if recently observed atmospheric circulation had occurred under preindustrial conditions. Recognizing this potential bias, our results likely represent an aggressive estimate of the contribution of sea-surface conditions to recent GrIS surface mass loss. Even so, this analysis reveals a minimal influence. Not only does this support previous work showing low SMB
sensitivity to adjacent sea-sea surface conditions due to the barrier to onshore advection from the marine layer presented by consistent katabatic outflow over the ice sheet (Hanna et al., 2009, 2014; Noël et al., 2014), but it also shows that any bias due to our treatment of sea-surface conditions likely had a negligible impact on our estimate of the total local thermodynamic contribution to recent surface mass loss. It should also be noted that while this study quantified the direct contribution of changes in local thermodynamic and sea-surface conditions to surface mass loss, it has been posited that these changes may
also contribute to GrIS mass loss indirectly by promoting the observed shift to more persistent atmospheric circulation patterns over Greenland. For example, several theoretical frameworks predict that persistent circulation states may become more common during summer under Arctic amplification (Coumou et al., 2014, 2018; Francis and Vavrus, 2012), and previous work has linked persistent ridging over Greenland to reductions in sea ice (Liu et al., 2016; Screen, 2013; Wu et al., 2013) and North American snow cover extent (Preece et al., 2023b).




Contrary to the hypothesis put forth by Stroeve *et al.* (2017) that higher SST and lower SIC may promote earlier melt onset in the spring which acts to precondition the ice sheet for later melt, the evidence here suggests very limited impact of sea-surface conditions on recent GrIS surface melt volume and no significant impact on seasonal melt timing (Fig. 4 and 10). In fact, the little impact by sea-surface conditions that does occur is maximized later in the melt season when ocean-atmosphere heat

exchange is greater than during spring (Fig. 6, 7, 9) (Screen, 2017). However, when considering the change in atmospheric thermodynamic fields (i.e., PGW1), these results show an advance in melt onset and an even greater delay in freeze onset in response to a warming atmosphere that has led to an overall lengthening of the melt season (Fig. 10).

Because MAR assumes a fixed ice sheet geometry, the results presented herein strictly describe the thermodynamic influence

on the SMB of the ice sheet; however, GrIS surface runoff and solid ice dynamics are not independent. Strong pulses of meltwater can cause rapid drainage through moulins that overwhelms the subglacial drainage network (Chu, 2014; Schoof, 2010). The consequent buildup of pressure increases basal sliding, causing a surge in ice velocity that increases glacial discharge and accelerates ice sheet thinning (Andersen et al., 2011; Chu, 2014; Schoof, 2010). Thus, it is likely that the thermodynamic influence on GrIS surface melt documented here has indirectly contributed further to sea-level rise via its

impact on ice sheet dynamics. Regardless, these results demonstrate that while the shift in summer atmospheric circulation over Greenland has been key to the acceleration of runoff from the GrIS, the change in the background thermodynamic state under Arctic amplification has markedly enhanced the observed surface mass loss beyond that which would have occurred if not for anthropogenic climate change.

**Code and Data Availability**

MAR data from this study are available through the Arctic Data Center (Preece et al., 2023a). ERA5 reanalysis data used to force the model can be accessed through the Copernicus Climate Data Store (Copernicus Climate Change Service, 2018). CESM-LE data used adjust the boundary conditions are available through the National Science Foundation (NSF) National Center for Atmospheric Research (NCAR) Research Data Archive (Kay et al., 2021). Merged Hadley-OI SIC and SST fields are hosted through Zenodo (Hurrell et al., 2020).

**Author Contribution**

JP, TM, and PA conceptualized the study. JP, PA, and GK designed the model experiments. JP, PA, and XF performed the model simulations. TM and MT led the project administration and funding acquisition. MT and PA supplied the computing resources. JP performed the formal analysis and prepared the manuscript in consultation with all co-authors.



## Competing Interests

At least one of the (co-)authors is a member of the editorial board of The Cryosphere.

## Acknowledgements

Computing resources to perform the MAR simulations were provided by the Lamont-Doherty Earth Observatory.

## Financial Support

This work was supported by NSF Arctic Systems Science award number 1900324, Strategic Environmental Research and
Development Program project number RC18-1658, NASA award 80NSSC17K0351 and Heising Simons Foundation award #
HSFOUNF 2019 - 1160. G.J.K. acknowledges support from the U.S. Department of Energy (DOE) Regional and Global Model
Analysis (RGMA) Program (DE-SC0021209).

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
