# Peer review of "Estimating the Thermodynamic Contribution to Recent Greenland Ice Sheet Surface Mass Loss"

_EGUsphere, 2025_

## Author Comment (AC1)

RC1: 'Comment on egusphere-2025-4140', Anonymous Referee #1, 03 Nov 2025

General remarks

This is an excellent and novel quantitative analysis of the contribution of the local change in background thermodynamic contribution to the Greenland Ice Sheet surface mass balance loss. It effectively separates out the role of sea-surface temperature and sea-ice concentration forcing, which is found (corroborating some but not all previous studies) to be relatively minor. A valuable analysis of the extreme melt case studies of 2012 and 2019, and the difference in terms of climatic forcing between these events, is provided. The analysis is thorough and the paper is clearly written. It will be of wide interest to Greenland climate scientists and ice-sheet specialists. I recommend publication following a minor revision addressing the points below.

***We thank the reviewer for their encouraging feedback. Please see our response (in bold) to each of your specific comments below.***

Specific comments

Line 26: please add the following recent relevant references:

Otosaka et al. (2023) https://essd.copernicus.org/articles/15/1597/2023/

Hanna et al. (2024) https://www.nature.com/articles/s43017-023-00509-7#publish-with-us

***Thank you for suggesting this recent work on the subject. These references will be added in support of the statement on Line 26.***

Figure 1: define PGW-1, PGW-2 etc. I the figure caption.

***Thank you for helping to make our discussion of the two model experiments clearer. We will also incorporate the suggestion from Reviewer #2 to rename the two PGW experiments to explicitly note the boundary fields that are altered in each experiment, and we will include those definitions in the Figure 1 caption.***

Lines 167-179: please add that while GCMs may capture the periodicity of internal climate variability they may not capture the magnitude of such variability. Will taking the CESM-LE ensemble mean be affected by signal-to-noise issues with the GCMs in capturing North Atlantic circulation change?

**We will edit the first sentence of the paragraph in question as follows to note that GCMs may not accurately resolve the magnitude of internal variability (changes italicized):**

> **"While GCMs aim to capture the periodicity of internal climate variability,** *they may not accurately resolve the magnitude of said variability and* **the precise timing of a particular mode of variability differs between individual ensemble members."**

**We hope that we are correct in interpreting the reviewer's mention of circulation change as referring to atmospheric circulation and not oceanic circulation. If the reviewer is referring to the failure of GCMs to capture the post-2000 change in North Atlantic**

atmospheric circulation, then that is precisely what motivated our study design. The use of ERA5 reanalysis in both the control and PGW simulations ensures that the synoptic-scale variability conveyed to MAR at its boundaries is the same in each of the model runs and is representative of the anomalous circulation observed between 2000 and 2019. So, we are not reliant upon the GCM representation of North Atlantic circulation in that sense. We only relied upon GCM output (i.e., CESM-LE) to provide an estimate of the long-term change in the background thermodynamic state of the atmosphere, which was then used to adjust the ERA5 boundary conditions in such a way that the anomalous circulation was preserved between the three model runs. We utilized the ensemble mean to remove some of the noise of internal variability and better isolate the long-term signal of the changing background state. The perturbation fields were computed from the long-term monthly mean for the preindustrial and control periods averaged over the 40 ensemble members of CESM-LE. In total, the monthly means employed in the computation of the perturbation fields represented 800 years of data for each period, which should effectively average out the internal variability.

In response to the suggestion from Reviewer 3, we conducted a comparison of average 500 hPa geopotential heights over Greenland to demonstrate this preservation of the large-scale circulation. We will add this comparison to the supplementary information and discuss it in section 2.

l.201: Are there issues with the accuracy of prescribed SIC and SST for 1880-1899 that may affected the method used?

Thank you for raising this issue. The Hadley-OI dataset does have certain limitations relative to ERA5 that should be mentioned, and these shortcomings also speak to some of the issues regarding model representation of ocean-atmosphere coupling raised by reviewers 2 and 3. Comparisons between gridded global SST products including the HadISST1 dataset that is the source of SST in the merged Hadley-OI product during the preindustrial period examined here generally show strong spatial and temporal agreement; however, the SST records provided by these datasets are least robust in areas that rely more heavily on spatial interpolation, such as along coastlines and near the sea ice front, where in situ measurements are less frequent. Furthermore, the relatively low spatial resolution of the preindustrial data struggles to capture finer-scale features such as boundary currents, as well as SST and its variability in coastal locations, and acts to smooth sharp SST gradients along ocean fronts (Yang et al., 2021; Hurrell et al., 2008; Hanna et al 2006). However, the disparities among datasets are generally much smaller than the long-term trends (Yang et al., 2021) and, therefore, should exert minimal influence on the signal that we seek to quantify in our analysis. Similarly, SIC from 1880-1899 is inferred from monthly, 1901-1930 climatologies of hand-drawn sea ice charts (Hurrell et al., 2008; Rayner et al., 2003) which should provide a reasonable estimate of the long-term change since the preindustrial period. However, these issues could have some impact on model representation of the local marine influence on Greenland Ice Sheet surface mass balance but are unlikely to have clear, systematic impact on the results for Greenland as a whole. We will include these points in a broader discussion on potential shortcomings in the model representation of ocean-atmosphere coupling in section 4 upon revision of the manuscript.

Hanna, E., Jónsson, T., Ólafsson, J., and Valdimarsson, H.: Icelandic Coastal Sea Surface Temperature Records Constructed: Putting the Pulse on Air–Sea–Climate Interactions in

the Northern North Atlantic. Part I: Comparison with HadISST1 Open-Ocean Surface Temperatures and Preliminary Analysis of Long-Term Patterns and Anomalies of SSTs around Iceland, Journal of Climate, 19, 5652–5666, https://doi.org/10.1175/JCLI3933.1, 2006.

Hurrell, J. W., Hack, J. J., Shea, D., Caron, J. M., and Rosinski, J.: A New Sea Surface Temperature and Sea Ice Boundary Dataset for the Community Atmosphere Model, Journal of Climate, 21, 5145–5153, https://doi.org/10.1175/2008JCLI2292.1, 2008.

Rayner, N. A., Parker, D. E., Horton, E. B., Folland, C. K., Alexander, L. V., Rowell, D. P., Kent, E. C., and Kaplan, A.: Global analyses of sea surface temperature, sea ice, and night marine air temperature since the late nineteenth century, J. Geophys. Res., 108, 2002JD002670, https://doi.org/10.1029/2002JD002670, 2003.

Yang, C., Leonelli, F. E., Marullo, S., Artale, V., Beggs, H., Nardelli, B. B., Chin, T. M., Toma, V. D., Good, S., Huang, B., Merchant, C. J., Sakurai, T., Santoleri, R., Vazquez-Cuervo, J., Zhang, H.-M., and Pisano, A.: Sea Surface Temperature Intercomparison in the Framework of the Copernicus Climate Change Service (C3S), Journal of Climate, 34, 5257–5283, https://doi.org/10.1175/JCLI-D-20-0793.1, 2021.

ll.252 & 480: please add the following highly relevant reference:

Hanna et al. (2021) https://rmets.onlinelibrary.wiley.com/doi/full/10.1002/joc.6771

*Thank you for highlighting this oversight. We will include this reference in support of the statements on lines 252 and 480.*

The green and blue lines on some plots look a bit similar; could they be distinguished more clearly?

*Thank you for raising this issue. We have included two samples below to illustrate how we will increase the contrast between the two time series throughout the manuscript.*

[Figure]

[Figure]

| Original Fig. 11 | Revised Fig 11 |

l.496: please add the following highly relevant reference to "more persistent circulation regimes under global warming":

Overland et al. (2012) https://agupubs.onlinelibrary.wiley.com/doi/full/10.1029/2012GL053268

**Thank you for highlighting this omission. We will add this reference in support of the statement ending on line 496.**